# Incremental Uncertainty-aware Performance Monitoring with Labeling Intervention

**Alexander Koebler**[1,2*]   **Thomas Decker**[1,3*]   **Ingo Thon**[1]
**Volker Tresp**[3,4]   **Florian Buettner**[2,5,6]
[1]Siemens AG   [2]Goethe University Frankfurt   [3]LMU Munich
[4]Munich Center for Machine Learning (MCML)
[5]German Cancer Research Center (DKFZ)   [6]German Cancer Consortium (DKTK)
{alexander.koebler, thomas.decker, ingo.thon}@siemens.com
volker.tresp@lmu.de, florian.buettner@dkfz.de

## Abstract

We study the problem of monitoring machine learning models under temporal distribution shifts, where circumstances change gradually over time, often leading to unnoticed yet significant declines in accuracy. We propose Incremental Uncertainty-aware Performance Monitoring (IUPM), a novel label-free method that estimates model performance by modeling time-dependent shifts using optimal transport. IUPM also quantifies uncertainty in performance estimates and introduces an active labeling strategy to reduce this uncertainty. We further showcase the benefits of IUPM on different datasets and simulated temporal shifts over existing baselines.

## 1   Introduction

Deployed machine learning models often face the critical challenge of distribution shifts, where the data encountered in production deviates from the data used during training. Many relevant shift scenarios involve changes over time, which are often gradual and continuous [26, 25]. These shifts are characterized by the fact that the statistical properties of the data or the environment change progressively rather than abruptly. This gradual nature can make time-dependent shifts more insidious, as they may not be immediately apparent but can still lead to substantial degradation in prediction quality over time [12]. Therefore, anticipating and understanding temporal performance changes is essential for ensuring the reliability and effectiveness of a machine learning model in dynamic environments [7]. However, directly monitoring the performance during the deployment is challenging as labeled data is often unavailable in production. Moreover, obtaining labels can be cumbersome, time-consuming, and costly, leading to delays in the assessment. Therefore, an increasing number of label-free estimation methods have been proposed that aim to anticipate the model performance purely based on unlabeled data available at runtime.

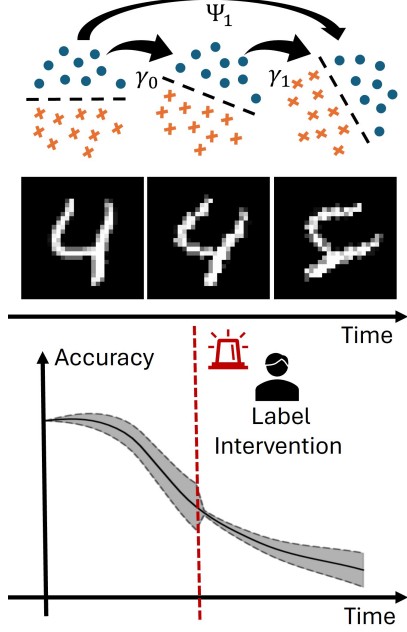

Figure 1: Illustration of Incremental Uncertainty-aware Performance Monitoring (IUPM) with label intervention.

---

[*]Equal contribution

Workshop on Bayesian Decision-making and Uncertainty, 38th Conference on Neural Information Processing Systems (NeurIPS 2024).

While this task is provably impossible in general [6], it can be approached by leveraging prior knowledge about the structure of the shift [13]. Existing techniques leverage strategies based on feature statistics [9], importance weighting [23, 3], model confidence [14, 13], disagreement between models [16, 1], or model differences after retraining [27]. However, only a few works explicitly try to incorporate the additional structure arising from the temporal nature of shifts into the estimation process [2, 28]. Furthermore, existing techniques are unable to quantify any uncertainty related to the performance estimate, cannot anticipate model degradation for arbitrary loss functions, and provide no information on how to best aid estimation with limited labeled data. To address these challenges we make the following contributions:

- We propose **Incremental Uncertainty-aware Performance Monitoring (IUPM)**, as a novel label-free performance estimation method tailored to temporal distribution shifts (see Figure 1).

- We quantify the uncertainty of the estimated performance to ensure reliable detection of model degradation during deployment.

- We introduce an active intervention step to reduce uncertainty in the performance estimate by labeling examples contributing the highest uncertainty to the performance estimate.

## 2  Incremental Uncertainty-aware Performance Monitoring (IUPM)

**Optimal Transport**    Optimal Transport (OT) aims at finding the cost-minimizing way to transform one probability measure into another [22]. Consider having $n_0$ samples from a domain $\Omega_0 = \{x_0^i\}_{i=1}^{n_0}$ and $n_1$ samples from another domain $\Omega_1 = \{x_1^i\}_{i=1}^{n_1}$ with corresponding empirical distributions

$$\hat{p}_0 = \sum_{x_0 \in \Omega_0} \frac{1}{n_0} \delta_{x_0} \qquad\qquad \hat{p}_1 = \sum_{x_1 \in \Omega_1} \frac{1}{n_1} \delta_{x_1}$$

where $\delta_x$ denotes the Dirac measure. For a cost function $c : \Omega_0 \times \Omega_1 \to \mathbb{R}^+$, the transformation of $\hat{p}_0$ into $\hat{p}_1$ can be formalized by a coupling $\gamma$ which represents a valid distributions over $(\Omega_0 \times \Omega_1)$ with marginals corresponding to $\hat{p}_0$ and $\hat{p}_1$. Identifying the cost-optimal coupling reads:

$$\hat{\gamma} = \arg\min_{\gamma \in \Gamma} \sum_{x_0 \in \Omega_0} \sum_{x_1 \in \Omega_1} c(x_0, x_1)\gamma(x_0, x_1) \quad \text{with} \quad \Gamma = \{\gamma \in \mathbb{R}^{n_0 \times n_1} \mid \gamma \mathbf{1}_{n_1} = \hat{p}_0, \gamma^T \mathbf{1}_{n_0} = \hat{p}_1\}$$

which can be solved using different algorithmic approaches [22]. In the discrete sample case the obtained $\gamma \in \mathbb{R}^{n_0 \times n_1}$ simply is a matrix with entries $\gamma(x_0, x_1)$. Moreover, the conditional coupling

$$\gamma(X_0 = x_0 | X_1 = x_1) = \frac{\gamma(x_0, x_1)}{\sum_{x_0 \in \Omega_0} \gamma(x_0, x_1)}$$

is a left-stochastic matrix whose entries can be interpreted as transition probabilities when moving from samples of $X_1$ to samples of $X_0$ following the most cost-efficient path.

**Incremental Performance Estimation using OT**    Consider a machine learning model $f$ that has been trained with labeled data $\{(X_0, Y_0)\}$ from the distribution $P_0(X_0, Y_0)$. Suppose $f$ is deployed in order to make predictions over time $t > 0$ with respect to data $\{(X_t, Y_t)\}_{t=1}^{T}$ each distributed with $P_t(X_t, Y_t)$. During runtime ($t > 0$) we only have access to unlabeled data from $X_t$ and our goal is to estimate only based on this information how well the model performs over time. To do so we proceed as follows: Let $\gamma_t(X_{t-1}|X_t)$ be the conditional coupling linking data from $X_t$ to data from $X_{t-1}$ in a cost-efficient way. Further, we define $\Psi_t(X_0|X_t) = \prod_{i=1}^{t} \gamma_i(X_{i-1}|X_i)$ describing the transition matrix obtained from composing all incremental transition matrices $\gamma_i(X_{i-1}|X_i)$ via matrix multiplication. It expresses the overall transition probabilities of going back to the labeled data available at $t = 0$ by connecting samples of two subsequent time points incrementally using an individual optimal transport coupling. Based on this we propose the following strategy to estimate missing labels for performance evaluation over time.

$$\hat{P}(Y_t | X_t) = \mathbb{E}_{\Psi_t(X_0|X_t)} [P(Y_0 | X_0)]$$

This means that our label estimate $\hat{P}(Y_t | X_t)$ arises as mixture distribution [10] combining labeled data in $X_0$ according to the accumulated incremental coupling results. This strategy is explicitly

reasonable for temporal shifts as we leverage their gradual nature by modeling subsequent distribution shifts incrementally using Optimal Transport. It implicitly assumes that the true decision boundary around data points from $X_t$ is similar to the ones of data points from $X_{t-1}$ that are linked via the cost-efficient coupling, which again suits in particular gradual shifts over time. Given an arbitrary loss function $\mathcal{L}$ to measure model performance and a set of samples $\Omega_t$ from $X_t$, the resulting performance estimate for IUPM at time $t$, denoted by $\hat{\mathcal{L}}_t^{IUPM}$, is given by:

$$\hat{\mathcal{L}}_t^{IUPM} = \mathbb{E}_{P(X_t)}\mathbb{E}_{\hat{P}(Y_t|X_t)}\left[\mathcal{L}(f(X_t), Y_t)\right] = \frac{1}{n_t}\sum_{x_t \in \Omega_t}\mathbb{E}_{\hat{P}(Y_t|X_t=x_t)}\left[\mathcal{L}(f(x_t), Y_t)\right]$$

Note that $\hat{P}(Y_t|X_t)$ is an actual predictive distribution that also internalized uncertainty for cases where linked samples have contradicting labels. Thus, we can use it to quantify the uncertainty of the anticipated performance using the expected standard deviation (SD) of the sample-wise loss estimates:

$$\mathcal{U}(\hat{\mathcal{L}}_t^{IUPM}) = \mathbb{E}_{P(X_t)}\mathrm{SD}_{\hat{P}(Y_t|X_t)}\left[\mathcal{L}(f(X_t), Y_t)\right]$$

**Labeling Intervention**   In addition to providing a means for users to consolidate their trust in the performance estimate, the quantified uncertainty $\mathcal{U}(\hat{\mathcal{L}}_t^{IUPM})$ can also be used to automatically trigger efficient relabeling when the uncertainty exceeds an acceptable level. To make the most efficient use of a limited labeling budget allowing to label only $m$ samples, we propose a Uncertainty Intervention (UI) strategy only querying ground truth labels for critical samples $x_t$ that contribute the largest uncertainty to the overall performance estimate:

$$\arg\text{top-}m_{x_t \in \Omega_t}\mathrm{SD}_{\hat{P}(Y_t|X_t=x_t)}\left[\mathcal{L}(f(x_t), Y_t)\right]$$

where $\arg\text{top-}m$ denotes the operator selecting the top $m$ elements maximizing the objective. The new labels are used to update $\Psi_t(X_0|X_t)$ such that $\hat{P}(Y_t|X_t = x_t)$ assigns a fixed label removing the accumulated uncertainty for sample $x_t$.

## 3   Experiments

We evaluate our proposed IUPM approach on two different settings and data modalities. First, to assess the general functionality of IUPM, we present results based on synthetic toy examples with continuous shifts in two-dimensional space. Second, to emphasize the connection to real-world scenarios, we perform experiments based on MNIST [18] and continuous image perturbations, e.g., resembling a gradual camera degradation. Throughout the experiments, we compare our approach to four existing performance estimation methods described in [13]. Average Confidence (AC) simply estimates the prediction accuracy as the expectation of the confidence for the predicted class across the data set. The more sophisticated Difference Of Confidence (DOC) [14] uses the discrepancy between the model confidence on the source and target data sets as an estimate of performance degradation. Average Threshold Confidence (ATC) [13] learns a threshold for model confidence on the initialization data set $\Omega_0$ and estimates accuracy on the current set as the fraction of examples where model confidence exceeds this threshold. Lastly, we consider Importance re-weighting as proposed by [3]. Additionally, we evaluate the use of the direct mapping $\gamma(X_0|X_t)$ for label transport and performance estimation inspired by [8]. We call this approach Non-Incremental Performance Estimation (NIPM). Across all experiments, we consider the model accuracy as the loss criteria $\mathcal{L}$ to evaluate performance. For more details, we also refer to Appendix A.1 and A.2.

**Translation and Rotation in Input Space** We use three two-dimensional toy data sets (Fig. 2) provided by [21]. For all three datasets, we train a Random Forest (RF), XGBoost (XGB), and a Multilayer Perceptron (MLP) classifier in the initial source distribution. After training at $t = 0$, all data sets are shifted for $t = 1, \ldots, 100$ steps to simulate gradual changes over time. For

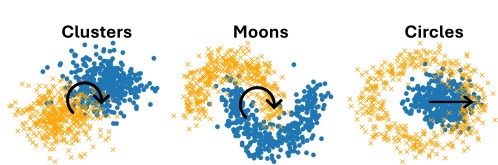

Figure 2: Synthetic two-dimensional toy datasets and corresponding shifts.

the "Clusters" and "Moons" data sets, this shift results from rotating both classes by $2°$ per step. The

Table 1: Mean Average Error (MAE) between ground truth and estimated accuracy using baseline methods and IUPM across three synthetic data sets and three different models.

| Method | Clusters | | | Moons | | | Circles | | |
|---|---|---|---|---|---|---|---|---|---|
| | RF | XGB | MLP | RF | XGB | MLP | RF | XGB | MLP |
| ATC | 0.4482 | 0.4701 | 0.4378 | 0.3821 | 0.3596 | 0.3696 | 0.3513 | 0.3455 | 0.3545 |
| AC | 0.4370 | 0.4908 | 0.4436 | 0.3264 | 0.3753 | 0.3532 | 0.2793 | 0.3342 | 0.3268 |
| DOC | 0.4417 | 0.4564 | 0.4522 | 0.3657 | 0.3501 | 0.3708 | 0.3594 | 0.3436 | 0.3495 |
| IM | 0.4580 | 0.4700 | 0.4746 | 0.3980 | 0.3609 | 0.3807 | 0.3572 | 0.3448 | 0.3508 |
| NIPM | 0.4303 | 0.4260 | 0.4731 | 0.2514 | 0.2256 | 0.2325 | 0.0822 | 0.0823 | 0.0793 |
| IUPM | 0.2852 | 0.2837 | 0.2996 | 0.0897 | 0.0837 | 0.1065 | 0.0330 | 0.0368 | 0.0335 |
| **IUPM$_{UI}$** | **0.0270** | **0.0272** | **0.0265** | **0.0244** | **0.0242** | **0.0222** | **0.0160** | **0.0157** | **0.0158** |

Table 2: Mean Average Error (MAE) between ground truth and estimated accuracy given random sample selection during intervention (RI) and our proposed Uncertainty Intervention (UI) across three synthetic data sets and three different models.

| Method | Clusters | | | Moons | | | Circles | | |
|---|---|---|---|---|---|---|---|---|---|
| | RF | XGB | MLP | RF | XGB | MLP | RF | XGB | MLP |
| IUPM | 0.2852 | 0.2837 | 0.2996 | 0.0897 | 0.0837 | 0.1065 | 0.0330 | 0.0368 | 0.0335 |
| **IUPM$_{UI}$** | **0.0270** | **0.0272** | **0.0265** | **0.0244** | 0.0242 | 0.0222 | **0.0160** | **0.0157** | **0.0158** |
| **IUPM$_{RI}$** | 0.0336 | 0.0327 | 0.0296 | 0.0256 | **0.0234** | **0.0198** | **0.0160** | 0.0177 | 0.0176 |

"Circles" dataset experiences a translation shift by $0.02$ in the x-direction only on the inner circle class. The results in Table 1 show that IUPM consistently outperforms all baselines across all datasets and models. The error in the accuracy estimate can be further reduced by intervening on the labels. In this case, we relabel the top 50% of $\Omega_t$ according to our introduced UI strategy once $\mathcal{U}(\hat{\mathcal{L}}_t^{IUPM}) > 0.1$.

**Comparison between Targeted and Random Label Intervention** To validate the utility of our proposed uncertainty indicator, we provide an additional experiment that quantifies the benefit of actively selecting the samples $x_k$ to be relabeled during label intervention. More Specifically, we compare relabeling $m$ samples based on our proposed instance-wise performance uncertainty $\arg \text{top-}m_{x_t \in \Omega_t} \text{SD}_{\hat{P}(Y_t|X_t=x_t)} [\mathcal{L}(f(x_t), Y_t)]$ with randomly selected $m$ samples from $\Omega_t$. The results in Table 2 show a considerable benefit of targeted sampling for the majority of scenarios, demonstrating that it can improve the efficiency of the labeling intervention under a limited labeling budget. This supports the hypothesis that correcting the labels of examples with the highest instance-wise uncertainty also has an increased benefit to the overall performance estimation.

Table 3: Mean Average Errors (MAE) between ground truth and estimated accuracy for a LeNet across three different shifts on the MNIST data set.

| Method | Rotation | Scaling | Translation |
|---|---|---|---|
| ATC | 0.3413 | 0.1715 | 0.3280 |
| AC | 0.4908 | 0.2284 | 0.3916 |
| DOC | 0.4695 | 0.2533 | 0.4165 |
| IM | 0.4303 | 0.6209 | 0.5701 |
| NIPM | 0.2025 | 0.0603 | 0.3150 |
| IUPM | 0.0833 | 0.0372 | 0.1177 |
| **IUPM$_{UI}$** | **0.0581** | **0.0355** | **0.0693** |

**Monitoring Performance Degradation due to Image Perturbations** To assess more complex shifts, we monitor a model classifying handwritten digits [18] that experience a shift caused by common image perturbations [20] such as rotation, used in a related context in [24], translation, and scaling of the digits. For this experiment, we trained a LeNet [19]. Further, we perform the matching

based on representations of the second classification layer of this LeNet, reducing the dimensional of the matched samples from $784$ to $84$ and demonstrating a generalizing approach to deal with higher dimensional data also used in [5]. All shifts are evaluated for $20$ steps, where Figure 3 illustrates the performance estimation for a 180° rotation. Even without intervention, IUPM can best estimate the performance. By allowing label intervention with the same settings as in the previous experiment, the performance estimation error can be reduced and the associated uncertainty can be limited to the predefined value $\mathcal{U}(\hat{\mathcal{L}}_t^{IUPM}) < 0.1$. The observations are further supported by the results on two additional shifts in Table 3.

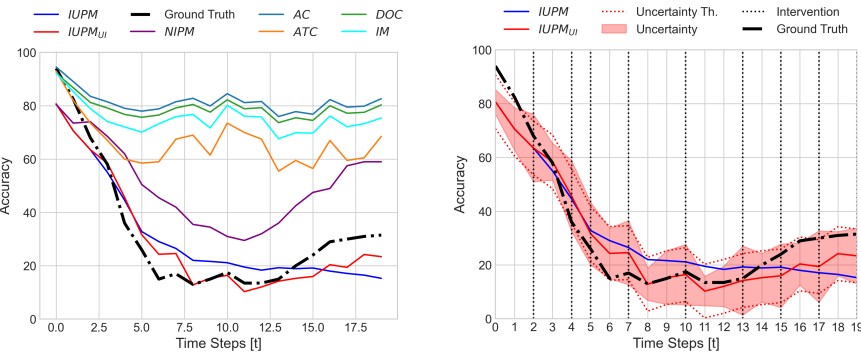

Figure 3: Performance estimation over time for the rotational shift on the MNIST digits accumulating to a 180° rotation after 20 steps. Left: IUPM is compared to the different baselines clearly illustrating that it offers the highest fidelity for the performance estimation. Right: IUPM with label interventions further increases estimation quality and limits the uncertainty to the predefined threshold.

## 4   Conclusion

We have introduced a novel method tailored to monitor a deployed machine learning model facing gradual distribution shifts over time. Our IUPM approach takes a step towards achieving a more reliable assessment of the quality of a model's output at run time by explicitly taking the uncertainty of the performance estimate into account, which current methods lack. This allows to simultaneous increase the user's confidence in the estimate and to perform a targeted label intervention to efficiently restore a sufficient trustworthiness of the system only when needed. We have shown that over two different data modalities and five different shifts, IUPM outperforms a number of existing performance estimation approaches. Incorporating additional insights from related fields such as gradual domain adaption [15] or active testing [17] as well as scaling to more complex data sets with real-world shifts, are interesting avenues for future research.

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

# A Appendix

## A.1 Baseline Methods

In this section we provide some additional information to the used baselines.
For the four existing baseline methods we relied on the implementations provided by [13]. Average Confidence (AC) simply estimates the prediction accuracy as the expectation of the confidence for the predicted class across the data set in step $t$ as $AC_{\Omega_t} = \mathbb{E}_{x \sim \Omega_t}[\max_{j \in \mathcal{Y}} f_j(x)]$. Difference Of Confidence (DOC) [14] uses the discrepancy between the model confidence on the source and target data sets as an estimate of performance degradation. To obtain an approximation of the performance in step $t$, the degradation is subtracted from the performance on the initialization data set $t = 0$
$DOC_{\Omega_t} = \mathbb{E}_{x,y \sim \Omega_0}[\arg\max_{j \in \mathcal{Y}} f_j(x) \neq y] + \mathbb{E}_{x \sim \Omega_t}[\max_{j \in \mathcal{Y}} f_j(x)] - \mathbb{E}_{x \sim \Omega_0}[\max_{j \in \mathcal{Y}} f_j(x)]$.

## A.2 Experimental Details

This section provides additional implementation details for the experiments.
For our IUPM implementation we rely on the entropic regularization optimal transport implementation with logarithmic Sinkhorn by [11].
As for the synthetic two-dimensional datasets, we used the data generator functionality provided by [21]. The "Clusters" data set is generated using the make_blobs function with a distance parameter of $1.0$. The "Moons" and "Circles" data sets are generated using the corresponding functions with a noise parameter of $0.2$ and a circle factor of $0.3$. For the training and initialization step, a training set of $800$ samples is generated, from which a validation and initialization set $\Omega_0$ of $200$ samples is partitioned. In each consecutive step, a set $\Omega_k$ with a different random seed is generated. We then apply a shift to the set corresponding to the step $k$, i.e. $k \cdot 2°$ for rotation and $k \cdot 0.02$ for translation. For the synthetic data, we use a Random Forest Classifier (RF) and a XGBoost Classifier (XGB) [4] with 50 estimators and a maximum depth of 5 as well as a Multilayer Perceptron (MLP) with a single hidden layer of size 128. For optimal transport matching, we use a regularization parameter of $10^{-4}$. Concerning the MNIST experiments, we adapt the image perturbation implementation introduced in [20] to the continuous setting. The used LeNet model has been trained for 100 epochs with early stopping based on a patience of 10 epochs and PyTorch's Adam optimizer with a batch size of 16 and a learning rate of $1e-3$. For the optimal transport matching, we use a regularization parameter of $1$. In this experiment, $\Omega_0$ also consists of 200 validation samples.

