# OpenReview forum: "Incremental Uncertainty-aware Performance Monitoring with Labeling Intervention"
_NeurIPS.cc/2024/Workshop/BDU — NeurIPS BDU Workshop 2024 Poster_

### Official Review · Reviewer_HDkb · 2024-09-26
**Temporal Shift Monitoring , with room for sensitivity checks and broader application**

**Rating:** 9
**Confidence:** 3

**Review:**

__Summary:__ The authors propose a way to monitor machine learning model performance over time, without needing a lot of labeled data. This method addresses the challenge of performance monitoring of deployed models, under temporal distribution shifts. Their system, Incremental Uncertainty-aware Performance Monitoring (IUPM), quantifies uncertainty of and queries labels only when absolutely needed.

__Quality:__
The proposed approach  IUPM is technically sound. The experiments are well-executed, with clear baselines, but could be expanded for more diverse real-world datasets. In particular, the method seems to rely heavily on the relationship between the original training data and the shifted data. If the original model was not trained on a representative sample, the system’s ability to estimate future performance could be compromised.

__Clarity:__
The paper is mostly clear but can be dense in places, particularly where technical details like OT and incremental performance estimation are discussed.

__Originality:__
The authors leverage optimal transport in an innovative way. The method is novel in its combination of performance estimation and uncertainty quantification for temporal shifts.

__Significance:__
The approach addresses performance degradation over time (a common and often overlooked issue in deployed models). The idea of selectively triggering label interventions based on uncertainty is also a practical contribution. IUPM has the potential to make an impact in fields where retraining or labeling is expensive, though its real-world implications could be explored further.  The method is designed for cases where shifts happen over time. It may not be well-suited to non-temporal shifts or contexts where the primary changes are driven by other factors (i.e., spatial, categorical…).

__Pros:__
- Innovative use of optimal transport
- Combines uncertainty quantification with selective labeling to reduce monitoring costs.
- Empirical results show improvements over existing baselines.
- Tackles a practical problem in model deployment.

__Cons:__
- The method assumes gradual distribution shifts, which may limit its application in scenarios with abrupt or unpredictable shifts
- Limited exploration of real-world, high-dimensional datasets beyond MNIST.  There is limited evidence of how the system performs in more complex, high-dimensional, real-world data with unpredictable shifts.
- Potential computational overhead for large-scale or real-time applications.

__High-level Feedback:__
The authors have developed a creative and promising approach to an important challenge in machine learning. The combination of performance estimation with uncertainty quantification and selective label intervention is both novel and practical. With some refinement in presentation and broader testing, this work could have a meaningful impact on the field. Expanding real-world use cases and simplifying some technical explanations will further enhance the paper’s accessibility and significance.

__Suggested Edits:__
Add more discussion on the implications of the experimental results. Why does IUPM outperform existing methods? Are there particular scenarios where it struggles? This kind of analysis will strengthen the contribution of the paper.

---

### Official Review · Reviewer_895K · 2024-09-26
**A novel approach to label-free performance estimation over temporal shifts**

**Rating:** 4
**Confidence:** 2

**Review:**

This work deals with an important challenge faced by real-world ML models. It proposes a label-free method for monitoring the performance of ML models after deployment when data distribution gradually shift from the training set over time. The use of optimal transport appears to be innovative and cost-efficient. Overall, it is an interesting paper but I am unsure of the real-world implications given the assumption of gradual shift and the simple datasets used in the experiments. See below for some details.

1. It would be useful to show to what extent IUPM works with abrupt changes in the dataset rather than a gradual shift, and what the boundary cases are (i.e. quantifying how "gradual" the shift needs to be for IUPM to work above baseline).

2. Does it work with higher dimensional data? The authors show an experiment with MNIST but real-world problems are much more complex. What are the computational complexity and scalability of IUMP? It would be useful to see more discussions on this.

3. Lines 129-132, "For this experiment, we perform the matching based on representations of the second classification layer of the LeNet, reducing the dimensional of the matched samples from 784 to 84 and demonstrating a generalizing approach to deal with higher dimensional data also used in [5]." Would this also work if matching is done on a later layer? It might be interesting to see a more comprehensive experiment on the trade-offs between matching on different layers and real-world guidance on choosing a layer.

---

### Decision · Program_Chairs · 2024-10-09

Accept (Poster)